# Blood Clot Phenotyping by Rheometry: Platelets and Fibrinogen Chemistry Affect Stress-Softening and -Stiffening at Large Oscillation Amplitude

**DOI:** 10.3390/molecules25173890

**Published:** 2020-08-26

**Authors:** Ursula Windberger, Jörg Läuger

**Affiliations:** 1Department for Biomedical Research, Decentralized Biomedical Facilities, Medical University Vienna, Borschkegasse 8a, 1090 Vienna, Austria; 2Anton Paar Germany GmbH, Helmuth-Hirth-Strasse 6, 73760 Ostfildern, Germany

**Keywords:** Blood clot, platelets, fibrinogen, rheometry, large amplitude oscillation, clot phenotype, clot compliance, nonlinear behavior, LAOStress, fibrin gel

## Abstract

(1) Background: Together with treatment protocols, viscoelastic tests are widely used for patient care. Measuring at broader ranges of deformation than currently done will add information on a clot’s mechanical phenotype because fibrin networks follow different stretching regimes, and blood flow compels clots into a dynamic non-linear response. (2) Methods: To characterize the influence of platelets on the network level, a stress amplitude sweep test (LAOStress) was applied to clots from native plasma with five platelet concentrations. Five species were used to validate the protocol (human, cow, pig, rat, horse). By Lissajous plots the oscillation cycle for each stress level was analyzed. (3) Results: Cyclic stress loading generates a characteristic strain response that scales with the platelet quantity at low stress, and that is independent from the platelet count at high shear stress. This general behavior is valid in the animal models except cow. Here, the specific fibrinogen chemistry induces a stiffer network and a variant high stress response. (4) Conclusions: The protocol provides several thresholds to connect the softening and stiffening behavior of clots with the applied shear stress. This points to the reversible part of deformation, and thus opens a new route to describe a blood clot’s phenotype.

## 1. Introduction

Thromboembolic disease is a leading cause of mortality worldwide [1]. While arterial thrombosis like myocardial infarction and stroke is associated with atherosclerotic plaque formation and occurs at high and oscillating wall shear stresses, venous thrombosis occurs predominately at sites of low blood flow and is associated with plasmatic hypercoagulability due to systemic inflammation or cancer [2]. The different pathophysiology of arterial and venous thromboembolism leads to different thrombus composition, depending also on the availability and qualitative properties of the cellular blood components.

In arteries, red blood cells (RBCs) migrate to the centerline whereas platelets migrate towards the vessel wall, where they bind to the von Willebrand factor (vWF)-collagen cluster on top of a vessel lesion [3] or a ruptured atherosclerotic plaque. This process is supported by the irregular shape of platelets (which creates a surface that can be subjected to tangential shearing forces), by platelets sensing their mechanical environment [4], and by the linear conformation of vWF in high shear flow. This is also supported by RBCs pushing the platelets to aggregate closer together [5,6]. Other platelets that come along with the blood flow bind to the nascent platelet layer via their α_IIb_β_3_ integrins, either by direct cell–cell contact, or mediated by fibrin(ogen) bridges. The circulating platelets are activated by, e.g., serotonin, thromboxane A_2_ and adenosine diphosphate (ADP) released from adhering platelets [5,7], and form a plug with distinct hierarchy [8,9]. Much of this plug contains tightly aggregated platelets within a fibrin fiber meshwork of varying density. There can be areas of less activated platelets around this core as well as intervening transition regions [7]. Numerical simulation has shown that outer parts of the clots can be detached by the RBCs that shear along the plug surface [5], whereas the dense core remains stable. The dense packing confines the enzymes of plasmatic coagulation, so that the clot can further densify, but at the same time it impedes the access of other platelets to the core, which limits clot size and stops clot growth [3].

In contrast, venous blood clots are formed in a low flow environment, where it is easier to capture platelets from the bloodstream, which reduces the need for vWF-mediated binding. Platelets can bind directly to collagen via their α_2_β_1_ and GPVI integrin complex, which triggers an intracellular cascade to modulate GPIIb/IIIa into a form competent to bind fibrinogen. Blood cell types are distributed more randomly at low flow, and RBCs can form rouleaux if the shear stress is low enough. RBCs also can be sterically trapped and packed into clusters, which makes clots labile to hemodynamic alterations [10], RBCs attach to fibrinogen by surface receptors (CD47, β_3_), or are firmly bound to platelets by ICAM-4 interacting with the platelet GPIIb/IIIa integrin complex [11,12]. RBCs induce platelet aggregation by ADP release, and activate the plasmatic coagulation pathway through membrane-bound domains [13,14]. Therefore, venous clots are highly dynamic structures as well. Moreover, clots vary also in regard to their size: whereas venous clots can grow to several centimeters, arterial clots are small, because the adhesion and attachment of platelets is balanced by the high shear flow [8,15,16,17]. Clot architecture is thus highly influenced by the environment [9], and the incorporated cell types significantly influence clot density and stiffness.

While it is difficult to control this heterogeneity in the laboratory, the mechanical performance of functional parts of a clot can be well described. Regarding clinical care of surgical patients, viscoelastic point of care (POC) tests are widely used. In conjunction with appropriate treatment protocols based on cutoff values, they offer the possibility of targeted use of blood products and prothrombotic therapeutic substances [18,19,20,21]. Viscoelastic clot properties seem to also be important in a variety of diseases. Blood clots of patients with past myocardial infarction can be 50% stiffer compared to healthy controls [22], and patients undergoing hemodialysis exhibit a denser fibrous clot structure with thinner fibers [23]. In addition, blood clots of patients with ischemic stroke show lower contraction [24], and thrombi that dislodged from deep venous sites to the pulmonary vasculature are less compact than the originating clots [25]. Considerable knowledge has been gained from pure networks by rheometry; e.g., it has been shown how such systems strain-stiffen [26,27,28] and how incorporated cells modify this response [29].

Clot firmness, the main parameter of POC viscoelastic tests, is measured as a response to imposed deformation. However, only a single deformation amplitude is applied, and the full behavior of fibrin networks is not captured. Measuring clot firmness at a broad range of deformation would add important information on the network behavior because fibrin networks follow different stretching regimes that alternate with the applied strain [26,27,30,31,32]. Including the non-linear behavior into a test protocol also makes sense insofar as blood flow is dynamic in the body, which requires a dynamic mechanical response from the clots. In the arterial branch, pulse propagation and intravascular pressure induce rhythmical stretch to the vessel wall and clots inside of coronary arteries are subjected to the movement of the beating heart [33,34,35]. The blood flow and the circumferential tensile strain challenge the vascular wall and adherent clots. Turbulent flow at a site of constriction can increase the wall shear stress fourfold [36], and the shear flow there is so high that clots can even be remodeled mechanically [3,8]. Numerical simulations describe how fluid shear rates affect clot size [16,17], but the influence of rhythmic deformation is far less explored. A clot with a high elastic modulus might be brittle and break earlier or even abruptly at high dynamic stress loading. However, if clots could adapt to the prevailing forces, this must affect the behavior in a positive way. Such a non-linear behavior must originate from the fiber system [37], in which platelets can be integrated.

Different rheological techniques have been used to capture the non-linear behavior of fibrin gels and other biological networks. The superposition of constant stress with a small amplitude oscillatory stress probes the elasticity under a certain pre-stress. By increasing the pre-stress levels the stress response of the network is thus measured [26,38,39]. Since the amplitude of the oscillatory stress is small, the resulting elastic modulus is the linear behavior of a statically, non-linearly stretched network. In contrast, large amplitude oscillatory shear (LAOS) reveals the full non-linear behavior of viscoelastic materials by analyzing the intra-cycle properties during an oscillation cycle [40]. By applying a more stable control mechanism in the non-linear regime, LAOS testing is performed in most instances by controlling the strain as the sinusoidal input function and measuring the resulting stress function (LAOStrain). For shear stiffening materials like biological fiber networks, controlling the stress as sinusoidal input function (large amplitude sweep test, LAOStress) [41,42], and measuring the resulting strain function is not a challenge, as pointed out by Lamer and co-authors [31]. LAOStress is superior to LAOStrain because the applied stresses can be related to the hemodynamic forces that act on blood clots.

In this experimental study we studied the characteristic strain response of clots obtained from blood plasma that were exposed to cyclic stress loading. We provide a protocol suitable for clot phenotyping that includes several thresholds to connect the softening and stiffening behavior with the applied shear force.

## 2. Results

### 2.1. Clot Formation

Platelet count (PLT) and plasma fibrinogen concentration (FIB) of the platelet-rich plasma samples (PRP) are given in Table 1. PLT differed significantly between individuals due to the different values in the corresponding whole blood. Comparison between individuals had to consider this. Platelet-depleted plasma samples (PDP) were cell free. PLTs in the dilutions were calculated by dividing PLT of PRP with the dilution. As expected, addition of platelets generated a hypercoagulable condition with accelerated clot formation and increased clot stiffness (Figure 1). Kinetics showed species-specific differences. Cow plasma needed more time to form the clot, but clots were the stiffest. In contrast, pig and rat clots were quickly formed. Human and horse lay in-between these two extremes. The storage modulus (G′) at the plateau of the kinetic curve (G’_plateau_) at PLT between 100–200 G/L was for human samples: 564 ± 68 Pa; for the cow sample: 1650 Pa; for the horse sample: 674 Pa; for the pig sample: 549 Pa; and for the rat sample: 885 Pa. Human whole blood (WB) samples had a G’_plateau_-value of 210 ± 29 Pa. PLT of WB samples was 114 ± 74 G/L.

### 2.2. LAOStress Test

Amplitude sweep tests covered the linear (LVE) and the non-linear viscoelastic behavior of the clots. Figure 2 showcases Bowditch-Lissajous diagrams of strain (γ(ω)) versus stress (τ(ω)) during characteristic stages in the test. The minimum-strain compliance (J’_M_) and the large-strain compliance (J’_L_) were calculated out of these plots. The tangent modulus at zero instantaneous stress displays J’_M_, and the secant modulus at maximum stress displays J’_L_ [41,42]. High J’_M_ and J’_L_ values indicate a compliant clot, whereas low J’_M_ and J’_L_ values indicate a stiff clot. As long as the clot is deformed in the LVE regime, the Bowditch-Lissajous plots remain elliptic and therefore J’_M_ and J’_L_ values are identical. When cyclic loading leads to non-linearity, the J’_M_ and J’_L_ values become disparate and the stress softening ratio *R* (*R* = (J’_L_ − J’_M_)/J’_L_) deviates from zero. Negative *R*-values indicate intra-cycle stiffening. Figure 3, Figure 4 and Figure 5 explain the development of G´, γ, J’_M_, J’_L_, and *R* during the different regions in the test. Figure 6 compares human and cow clots by the shear stress spectra of G´, J’_M_, J’_L_, and *R*.

#### 2.2.1. Description of the Regions in the LAOStress Test

The values (G´, J’_M_, J’_L_, *R*) depended on the platelet concentration at low shear stresses, but not at high shear stresses. We postulate four test regions, shown also graphically in Figure 7, for cases with or without RBCs and/or platelets.

Region 1 shows the LVE behavior of the clot. G´, J’_M_, J’_L_ values were constant, *R* values were zero. G´ increased by power law with the platelet concentration in every individual (mean exponent: 0.30; *R* > 0.83).Region 2 comprises the shear stress range from the start of non-linearity up to the onset of macroscopic shear stiffening. G´, J’_M_, J’_L_, *R* were no longer constant. In platelet-containing samples as well as in WB clots, G´ decreased and both compliances increased (region 2a). The clot softened. The more platelets that were present, the longer was this region. In PDP samples the compliances either remained constant or decreased while G´ increased. PDP samples did not soften. At a certain shear stress the compliances diverged and region 2b began. This stress value (τ_D_) indicates the onset of irreversible processes due to cyclic stress loading. τ_D_ shifted to higher values with the platelet count. The stress-softening ratio (*R*) became negative from that point on (Figure 4). Softening continues since both compliances increased and G´ decreased further, but the decrease of *R* indicates that cyclic stress loading forced the structures into a stiffening response at the same time. The maximum of J´_L_ indicates the end of region 2b. The more platelets that were present, the longer was region 2b (reaching from τ_D_ to τ_L_; Figure 5). In the subsequent region 2c (from τ_L_ to τ_M_), stiffening dominated although the increase of J´_M_ still indicates softening processes. Region 2c ends at the J´_M_ maximum (τ_M_). τ_L_ and τ_M_ shifted to higher values with the platelet concentration and scaled by PLT^3/5^ (*R* > 0.90; see also the arrows in Figure 6c).Region 3 starts at τ_M_ and ends where yielding sets in. J’_M_, J’_L_, and *R* decreased, and G´ increased in parallel. Only stiffening occurred. The several J’_M_, J’_L_, and G´-curves from the different platelet concentrations of individuals narrowed and overlapped in many cases. Figure 2d shows the overlapping Bowditch-Lissajous plots of clots with 1:64 and 1:4 platelet titers from the human individual 01, demonstrating this identical behavior. The maximum G´-value did not vary with the platelet concentration (783 ± 40.3 Pa in 11 clots from the human individuals; Spearman r = − 0.21), nor did the final strain (125 ± 29%; Spearman r = − 0.16).Region 4 shows clot weakening due to macroscopic yielding. Most clots became compliant again; only few broke abruptly.

#### 2.2.2. Influence of the Species

Among the species tested, cow took an exceptional position. Cow clots featured the lowest compliances and the highest shear moduli throughout the whole test (see Figure 6). The stress required to undergo a behavioral change was also higher than in all other clots at similar platelet counts. In contrast, the strains were the lowest (Figure 8). Cow clots were unable to shear-stiffen like the clots of other species could do. J’_M_ did not turn down (which marks τ_M_ in other species) but only displayed a shoulder and increased further until the clots yielded. This means that softening processes were still ongoing. Horse, rat, and pig clots generally showed similar compliances to the clots from the human individuals although distinct species-specific differences concerning the stress thresholds were present, e.g., τ_D_ occurred earlier in the horse and later in the rat (Figure 8). The rat clot showed a J´_M_ shoulder instead of a maximum, similar to the cow clot, confirming stiffness [43]. The G´-, J´_M_/J´_L_-, and *R*-spectra of the species are provided in the Appendix A.

#### 2.2.3. Results Summary

G´, J´_M_, J´_L_, and *R* in regions 1 and 2, as well as τ_D_, τ_L_, and τ_M_ shifted dose-dependently with the platelet concentration. When τ_M_ was passed, the clot dynamics became independent from the platelet count, except in cows. Clots from more diluted samples could be strained more easily but gained the equivalent G´-modulus and strain at maximum stretch-out compared with the clots from more concentrated samples. Figure 9 illustrates that the platelet count determines the shape of the compliance curves. Figure 10 shows that clots lose the ability to shear-stiffen and also break abruptly without preceding yielding when the platelet count is too high.

## 3. Discussion

Platelets are firmly incorporated into the fiber network through covalent binding of the distal end of fibrinogen to the activated α_IIb_β_3_ platelet integrin [44,45]. More central sites on the α-chain of fibrinogen also engage with platelet receptors and position the molecule in a side-to-side mode, assuring a distance of about 10–20 nm between two platelets [46]. Platelets influence the coagulation pathways, activate other platelets, and actively influence fibrin binding [47], but they do not interfere with fibrin polymerization. Fiber formation starts with the self-assembly of fibrin molecules into a linearly staggered conformation with a 22.5 nm periodicity, following thrombin-mediated cleavage of fibrinopeptides A (and later also B) from the central region of the molecule. In subsequent steps the two-stranded oligomers grow longitudinally to protofibrils, assemble laterally to fibers, and become covalently fixed by end-to-end γ-chain-, side-to-side α-chain-, and α-γ-chain-crosslinking of monomers under the influence of FXIII and fibronectin [48,49]. Fibrin is organized in parallel mode between the filopods of activated platelets [50]. The filopods extend along the fibrin fiber, pull the fiber, and eventually densify fibrin into bundles and clusters [51,52,53]. The network shrinks with time due to the contractile forces of platelets [54,55], which require strong platelet-fiber bonds. Platelets include a set of branching points into the network, which is inhomogeneous in respect to size and the number of fibers radiating out of them, but which always offers good coupling opportunities for fibrin. Together with adherent fibrin mass they generate stiff micron-scale building blocks in the fibrous meshwork.

In our experiments, platelets stiffened the clots and modified their response to low, intermediate, and high shear forces in a specific way. Opposing behaviors like softening and stiffening took place at the same time, which made the response more complex when platelets were added. The gain of stiffness with platelet addition reflects the increased contacts between the fibers and the contractile force of platelets [56]. It is very unlikely that the 2–4-microns-sized platelets would increase the elastic modulus of the clot alone through their passive presence.

Clots often but not always gain stiffness when cells are added. G´ is generally higher in whole blood clots compared to PDP clots. This was found in a numerical simulation [57], in experiments [32,58], and also in this study. One might think that RBCs increase elasticity by adding compact matter (apparent Young´s modulus of single RBCs is in the range of kPa [59]). However, the higher modulus of whole blood clots is the result of the simultaneous presence of platelets. In in-vitro tests, G´ is usually lowered when the hematocrit is raised [60]. This is due to the fact that RBCs lower the available plasma volume and therefore the fibrinogen content in the finite sample volume. RBCs also withdraw space for fiber propagation and prevent the alignment of the fibers to parallel bundles when the clot is stretched. This explains the blunted non-linear response. A similar effect was observed when fibroblasts were kinetically entrapped within the network [29]. It therefore depends on the circumstance if the added cells can be an integral part of the fiber system or if they represent fiber exclusion zones. The same may apply for other blood components like proteins or lipids. Compounds that do not become a functional or integral part of the fiber meshwork but lie passively in its voids must be recognized as confounding factors for the mechanical performance of the system. Although RBCs can be firmly attached to platelets [12] and incorporated into the meshwork, most of them will be passively locked in, where they not only hinder fiber bending, but also start to roll or deform when the clot is sheared. This can prevent the transmission of forces to the fibers. In addition to RBCs, platelet clusters might deform as well and would then add compliance to the network; e.g., we observed that clots with more platelets gained slightly higher strains between 200 and 800 Pa sinusoidal shear stress input (compare with Figure 3a), presuming the evolution of minute residual strains (γ_res_) in test region 3. Note that the highest strains (at >800 Pa) were independent from the platelet count and that the Bowditch-Lissajous plots overlapped there (Figure 2). Sound γ_res_ typically develop when ramps of constant shear stresses are applied in order to slowly stretch and relax fibrin gels [61]. By following this protocol, we also found substantial γ_res_ in human PDP and PRP clots, with the PRP clot exhibiting the lower γ_res_ (Appendix A). γ_res_ indicates the degree of irreversibility that is certainly present in all samples exposed to external loading. However, since residual strains develop quickly and we discard the first four loading cycles to read only the fifth one, we expect to probe a steady-state system that has already adapted to the higher amplitude. This also includes the hypothesis that we get access to that part of the material that deforms reversibly, although some very minor impreciseness cannot be excluded within a certain shear stress range.

In region 2, G´-curves narrowed systematically towards each other until they overlapped. Clots with low platelet count started with a low G´ but yielded at similar G´-values like clots with high platelet count that started with a high G´. The ability to stiffen was therefore reduced when platelets were concentrated, but the final stiffness remained unaffected (Figure 6). The explanation for the reduced stiffening of platelet-enhanced samples lies in concurring processes. Fibrin gels with low monomer concentration in regard to physiological conditions typically show strain hardening [26,30]. Fibrin gels with higher monomer concentrations were found to soften within a certain strain range [62]. In our study, J´_M_, J´_L_, and *R* indicate that the material stiffened and softened at the same time. Concurring processes are typical features of anisotropy, and platelets augment them due to their good correlation to the fibers. Their quantity determines how far a clot must undergo softening before stiffening can dominate. In other words, platelets prolong the shear stress range for softening processes as shown by the shift of the thresholds (τ_D_, τ_L_, τ_M_) to higher shear stress. Platelets that are firmly incorporated into the network reduce the mesh size and stress the fibers. Since the distances between the platelet clusters and the condition of the fiber bundles are unequal on a microscopic scale, the clusters experience non-affine displacement during LAOS. The irregular local stresses cause different fiber deformations. Some will buckle or bend, which makes the network compliant [63], whereas others will stretch and contribute to elasticity. Such a process was also suggested for fibroblast-enhanced fibrin and collagen gels [64,65].

As the shear stress increases, more fiber bundles can stretch out and contribute to elasticity until all fibers are oriented in the direction of the drag to uptake the load equally, and to reduce the local stress [66]. It was shown previously that the elasticity of pure fibrin gels as well as fibrin gels seeded with fibroblasts became independent from the fibrinogen concentration when the strain exceeded a critical value [26,28,29,30]. We observed a similar behavior. In all clots from an individual (including also the animal species, except the cow), the curves of G´, γ, J´_M_, J´_L_, and *R* fairly collapsed into one curve. Although the stress threshold shifted to higher values when platelets became concentrated in the samples, at ≈ 300 Pa all curves became parallel and remained so until the network yielded macroscopically. The threshold at which sole network stiffening sets in is given by τ_M,_. If parallel alignment of structures and affine deformation start at this point, it requires partial decoupling of fibers from the branching, which is easily attained when fibers are only entangled, but platelets bind several fibers covalently in a star-like assembly [52]. If these bonds break, the clot would not stiffen but would yield as soon as the distance between the links becomes longer than the contour length of the filaments in-between. How can such a highly anisotropic network align its fibers into a parallel geometry to stiffen as a whole? It is only possible until a certain platelet concentration. When there were too many platelets the curves did not unify. Clots from PRP broke apart while they were still in the alignment regime (Figure 10). However, when the fibers were too stiff, the values did not unify to one master curve, either. This was observed in the cow.

The amino acid sequence and secondary structure of bovine fibrinogen was investigated to some degree for the γ- and α-chains. Two disulphide bonds between the γ8 and γ9´ and the γ8´ and γ9 residues that are not present in the human analogue [48] link the two abutting γ-chains in the central region [67]. The asymmetric γN-domain is therefore covalently fixated in bovine fibrinogen. It was further shown that the γ-domain receptor pockets face the same direction on the two strands of a protofibril, which was proposed to influence their twisting [68]. The α-chain is shorter, containing fewer 13-residues repeats in the central region [69]. Since these repeats are supposed to behave as spring-like tethers for the αC-domain, the authors expect that the reduced length makes the αC-polymer stiffer. A compact structure in the dangling αC-domain was suggested by calorimetry in a further study [70].

In our experiments cow clots needed a long time to form and their moduli were doubled compared to human clots. The strains were reduced, indicative of a stiff clot. The extended time to stabilize the clot points to processes involving α–α-crosslinks [70], and the lower molecular weight of the α-chain [71] together with the lower number of extensible repeats would also indicate that the stiffness involves the αC-polymer. In addition, supercoiling of protofibrils must be affected by the conformation of the γ-chain, which reduces their longitudinal elasticity. Whatever the underlying cause on a molecular level is, bovine clots are stabilized. The clots needed more stress to undergo a behavioral change, were less compliant than other clots throughout the whole test, and responded to cyclic stress loading with a smaller compliance change (note the similar platelet concentrations used in Figure 8). In addition, the clots were unable to stiffen without also showing overlaying processes that softened them at the same time. This suggests a high resistance against fiber bending. J´_M_ did not turn down but showed only a tentative local maximum after which it continued to increase (see Figure 6d). Possibly, the fiber system is too rigid to compensate for the stiffness of the platelet-fiber clusters. Although affine deformation ameliorates after this tentative maximum, the load will never be distributed equally inside the clot, and it is surprising that the meshwork could sustain a shear stress of up to ≈ 2 MPa.

In summary, when physiological fibrin networks of native monomer concentrations are sheared in vitro, platelets modify the response in a dose-dependent manner by shifting the shear stresses that are needed for a behavioral change by power law to higher values. Platelets make the clots stiffer at lower stress levels and reduce their compliance. This is obviously due to the fact that platelets include foci of high connectivity to the network and drag on the fibers to which they are bound. According to the non-affinity model [72], platelets also bring inhomogeneity into the fiber network that leads to spatially correlated deformations and shifts the degree of non-affinity to higher stresses. After a critical shear stress threshold given by the J´_M_-maximum, the behavior becomes independent from the platelet quantity in most fibrin networks. Even if a clot in the vasculature had an irregular platelet distribution, it should thus be able to stretch out fully as long as the surrounding fiber network can compensate the local resistance. We like to hypothesize that high compliance could improve the “pumping” of solutes through the network when the clot is exposed to rhythmic cycles of deformation and relaxation. The clot would then “breathe” with the pulsations to facilitate network remodeling. We note that our approach is based on the performance of fiber networks and not platelet aggregates. Shear stiffening originates from the fiber system that is modified when platelets are present as well, and it is prevented when the platelet concentration is too high (compare with Figure 10). The bovine networks show that shear stiffening is also reduced when the fibers are too rigid, even when the platelet concentration is very low. Since the fiber meshwork can uptake the equivalent loads at a series of low and higher platelet counts, a high platelet count is not mandatory to enhance the breakup stress. In contrast, any addition of platelets beyond an individual threshold, with e.g., the aim to enhance clot stiffness, would at the same time make parts of the clot brittle because it blocks the stiffening response of the fiber system.

Several factors influence the architecture and the stiffness of fibrin clots [48,73], which were not controlled in the present study. For example, the use of native blood plasma containing numerous proteins also having posttranslational modifications changes the matrix in which the fiber meshwork is embedded and can modify effects that might be more obvious in fine networks [74]. However, a native sample is needed to describe the phenotype of a patient´s blood clot. Centrifugation must be performed with great care to avoid platelet activation prior to the test. First results on fibrin gels from PDP samples exposed to lipopolysaccharide (LPS) show that our approach is useful [58]. LPS-gels had a denser and less uniform fiber structure. This structure was accompanied by lower compliances, a greater degree of concurring processes (softening, stiffening), and a shift of macroscopic shear-stiffening (τ_M_) to higher shear stresses.

## 4. Materials and Methods

### 4.1. Blood Samples

Blood was obtained from healthy human volunteers (age: 23–33, BMI < 30, non-smokers, no intake of any medication or herbal remedy for the last 7 days) after giving informed consent. Blood collection was approved by the ethics committee of the Medical University Vienna, Austria (EK1371/2015). Study participants received a unique number that was used to guarantee anonymity, and investigations were carried out following the rules of the Declaration of Helsinki of 1975, revised in 2013. Blood withdrawal from horse, pig, rat, and cow was approved by the institutional ethics and animal welfare committee of the Veterinary University Vienna and the Medical University Vienna, Austria, and by the national authority according to Animal Experiments Act (BMWF-66.009/0284-WF/V/3b/2019 and BMBWF-68.205/0092-V/3b/2019). Blood was drawn using a Vacuette blood collection system (Greiner Bio-One, Kremsmünster Austria), containing 3.8% sodium citrate for anticoagulation.

### 4.2. Sample Preparation

Whole blood (WB) from the animals was kept between 15 and 20 °C in an insulated bag for a maximum of two hours during transportation to the laboratory. Human blood was processed 15 min after withdrawal. One portion of whole blood was removed to generate whole blood clots. Meanwhile, the remaining portions of the blood samples were centrifuged at 325 g for 8 min. The upper two thirds and the lower third of the plasma of each sample were collected separately and further processed. The upper two thirds were centrifuged at high speed (at 2310 g for 20 min). The superficial plasma layer was discarded and the central fraction was collected and labelled platelet-depleted plasma (PDP). The lower third of the plasma column was centrifuged at 260 g for 20 min, and its upper third was again discarded and only the central third was collected. This fraction was indicated as platelet-rich plasma (PRP). PRP was analyzed for platelet count (PLT, in G L^−1^) and fibrinogen concentration (FIB, in g L^−1^). PDP was controlled for quality using bright-field microscopy to detect the presence of any blood cells. PRP was joined with autologous PDP to dilute the plasma samples the way it is performed in serology (titers: 1:64, 1:32, 1:16, 1:8, 1:4), which were used subsequently for rheometry. In order to avoid any time delay and especially platelet activation by frequent mixing, PRP and PDP were not mixed to gain a defined platelet count, but to gain a certain titer. Titers were prepared by pipetting the exact amount of PDP into an eppendorf vial in which the respective amount of PRP was carefully suspended. As a result of this approach, the platelet counts in the titrated samples are different on an inter-individual basis (but all individual samples were titrated the same way). The hematological profile was performed with Sysmex XN-2000 (Sysmex, Tokyo Japan). Plasma fibrinogen concentration was measured by the method of Clauss.

### 4.3. Rheometry

A Physica MCR 301 rheometer (Anton Paar, Graz, Austria) was used. Temperature was Peltier controlled and set to 37 °C. A tempered hood mounted the measuring system and a silicon oil filled evaporation blocker prevented sample drying. The sand blasted stainless steel cone-plate measuring system (50 mm diameter, 1° cone angle, 0.1 mm tip truncation) was filled with 580 µL sample after re-calcification with 40 µL of 0.2 M CaCl_2_-solution (TEG^®^ Hemostasis System, Haemonetics, USA). To generate the clots in the cone-plate geometry, time sweeps were conducted at constant frequency (1 Hz) and low deformation amplitude (0.01%) to ensure only minimal mechanical interference with the ongoing clotting process [75,76]. The storage modulus (G´) was recorded to show the development of elasticity with time. Such kinetic tests were conducted until a G´-plateau value was reached, which indicates completed clotting. G´ was used to determine clot stiffness and was calculated from the stress (τ)/strain (γ) relationship by using the shift of the phase angle (δ): G´ = τ/γ cos(δ).

As soon as the G´-plateau value was reached, a stress amplitude sweep test (LAOStress) was started to investigate the dynamic behavior of the clot during deformation. At constant angular frequency (ω, 1 rad s^−1^), we set several logarithmic shear stress increments (τ(ω) from 1–5000 Pa) and measured the strain response (γ(ω)) of the clot. Since the first harmonic modulus can be a misleading measure during non-linear deformation, the oscillation cycle for e[ach stress level was analyzed by means of a recent model [40], using its extension to sinusoidal stress input functions [41,42]. To equilibrate time effects, we performed five oscillation cycles and took the last cycle for the LAOStress analysis.

### 4.4. Statistic

The stress spectra of the parameters (G´, γ(ω), J’_M_, J’_L_, *R*) were exported from the Rheocompass software (version 1.19) and processed in GraphPad Prism (version 8.1; GraphPad, San Diego, CA, USA) and Microsoft Excel for Mac 2011 (version 14.3.0) on MacOS Sierra. To test the effect of platelet count on clot stiffness, we used the plateau value of the kinetic test (G´_plateau_). To test the effect of platelet count on the clot´s non-linear behavior, we used the critical stress values that correspond to the J´-moduli´s divergence (τ_D_) and maxima (τ_L_, τ_M_). The relationship of G´_plateau_, τ_D_, τ_L_, and τ_M_ with the platelet count was determined by non-linear regression. Table 2 explains the parameters. The Spearman correlation coefficient was calculated to test if the G´ value before the clots yielded (G´_max_) and the corresponding strain value (γ(ω)_max_) depended on the platelet count. These correlations were done for each human individual separately. We did not perform technical repeats in order to avoid further time delays since several samples from one individual had to be tested on the day of sampling. Repeatability of our approach can be accessed from a recent study [58], where we used platelet-depleted samples.

## 5. Conclusions

We tested the dynamic response of clots generated from native plasma and we found good correlation of parameters with the platelet count in five mammalian species. The protocol allows the extrapolation of stress and strain values to identify the degree of softening and stiffening, which opens a new route to describe a blood clot phenotype. Once the clot is formed, the LAOStress test takes approximately 30 min, which makes it feasible for a clinical laboratory setting.

## Figures and Tables

**Figure 1 molecules-25-03890-f001:**
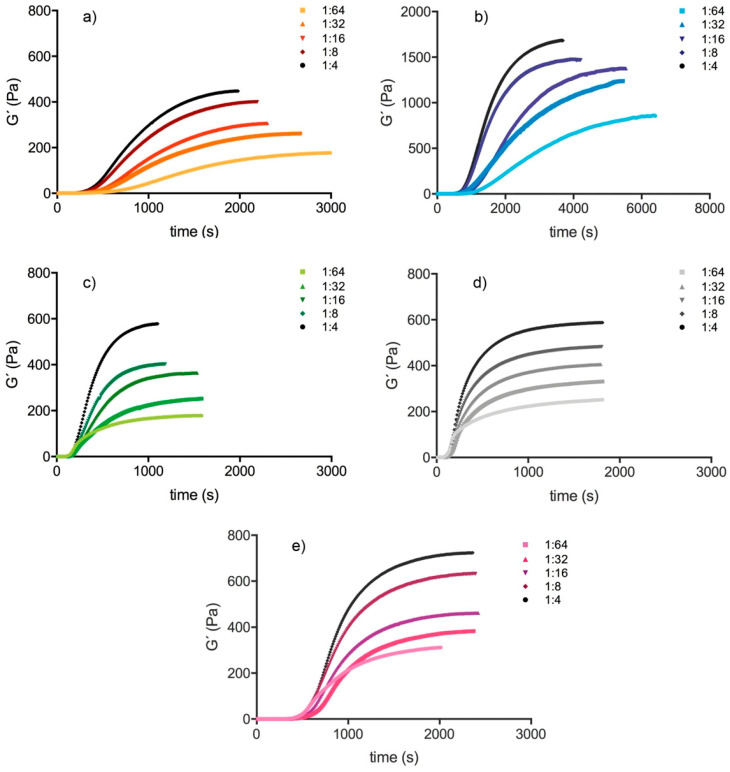
Kinetic of clot formation in (**a**) human (ID: 01), (**b**) cow, (**c**) pig, (**d**) rat, and (**e**) horse at different dilutions (e.g., 1:4 is the 1:4 dilution of PRP (platelet-rich plasma samples) in PDP (platelet-depleted plasma samples)).

**Figure 2 molecules-25-03890-f002:**
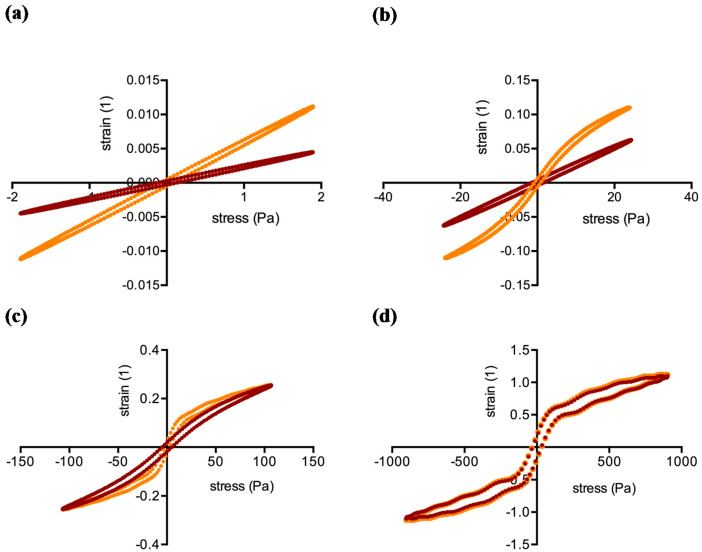
Bowditch-Lissajous plots of clots generated from human blood plasma (ID: 01) with a 1:64 (orange curves) and a 1:4 (brown curves) dilution of PRP in PDP. (**a**), Linear viscoelastic region; sample 1:64 experiences higher deformation. (**b**), At 24 Pa stress input, sample 1:64 responds nonlinearly, while sample 1:4 is still in its equilibrium. (**c**), At 106 Pa both samples respond nonlinearly; note that the sample with the higher platelet concentration (1:4) now gains the same strain as sample 1:64. (**d**), At 900 Pa both plots overlap (region 3).

**Figure 3 molecules-25-03890-f003:**
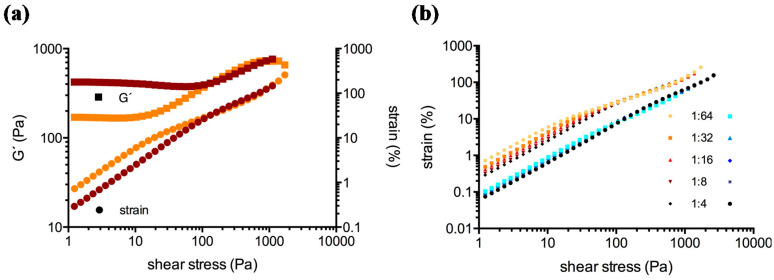
(**a**), Strain response to the applied stress of clots generated from human blood plasma (ID: 01) with a 1:64 (orange curves) and a 1:4 dilution (brown curves) of PRP in PDP. A clot with a lower G´-value at LVE (linear viscoelastic behavior, orange) allows higher strains than a clot with a higher G´-value (brown). Beyond 200 Pa the strain of the 1:4 dilution becomes slightly higher than its 1:64 counterpart. (**b**), Comparison of strains of human (orange curves) and cow clots (blue curves) during the test. There is very little difference in the strains of cow clots (note the log *y*-axis).

**Figure 4 molecules-25-03890-f004:**
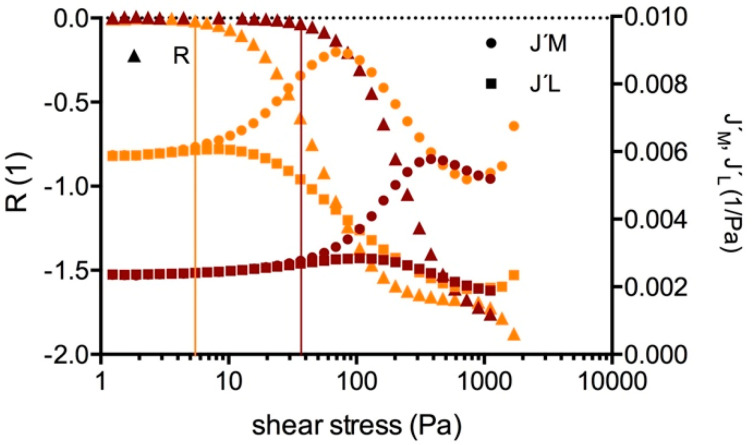
Compliances (J´_M_, J´_L_) in parallel with the stress softening ratio (*R*) in clots generated from human blood plasma (ID: 01) with a 1:64 (orange curves) and a 1:4 dilution (brown curves) of PRP in PDP. The vertical lines indicate the onset of *R* becoming negative (start of region 2b, τ_D_), indicating the start of intra-cycle processes.

**Figure 5 molecules-25-03890-f005:**
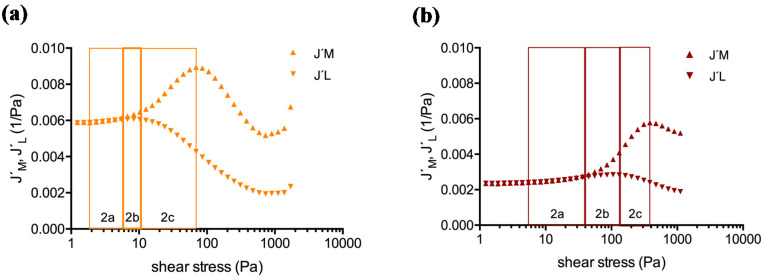
Clots generated from human blood plasma (ID: 01) with a (**a**) 1:64 and a (**b**) 1:4 dilution of PRP in PDP: three subregions of region 2. The more platelets are included into the network, the more pronounced is the softening. The critical shear stresses shift to higher values when platelets become concentrated.

**Figure 6 molecules-25-03890-f006:**
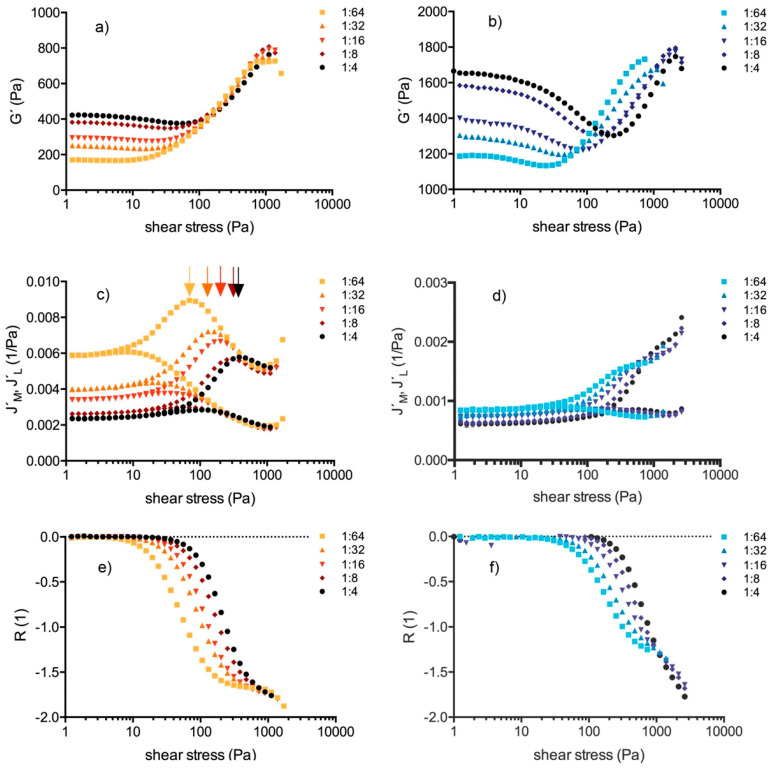
(**a**,**c**,**e**) Human (ID: 01) vs. (**b**,**d**,**f**) cow clots. Upper panel: storage modulus (G´) at LVE increases with the platelet concentration. In human clots, the G´ curves start to overlap at stresses > 200 Pa, whereas in cow clots the G´-curves become right-shifted. Central panel: compliances (J´_M_, J´_L_) increase with the reduction of platelets in the clots. The arrows (**c**) indicate τ_M_ (end of network alignment). Such a threshold cannot be defined in cow clots since the J´_M_-curves do not display a turning point. Lower panel: stress softening ratio (*R*) starts to deviate from zero at lower stress when fewer platelets were present. Curves are shifted to higher stresses in the cow.

**Figure 7 molecules-25-03890-f007:**
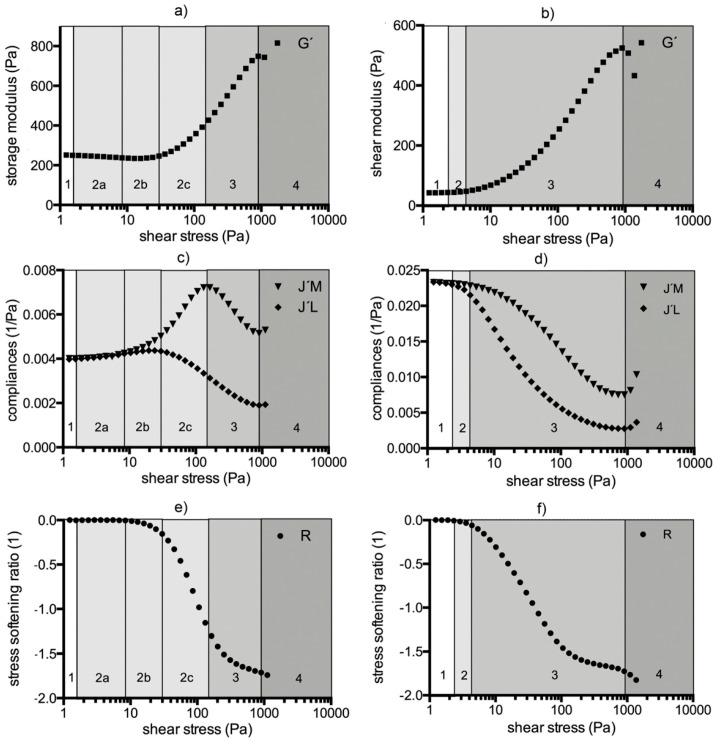
The LAOStress test comprises four regions. (**a,c,e**)**:** Test regions when blood cells (platelets and/or RBCs (red blood cells)) are present. Region 1: LVE region (G´ is constant, J’_M_ = J’_L_). Region 2 comprises three subregions. Subregion 2a: G´ decreases and both compliances (J’_L_ and J’_M_) increase but do not diverge: J’_L_ = J’_M_, *R* = 0; the clot softens. Subregion 2b: compliances diverge with J’_M_ > J’_L_ and both compliances increase; *R* starts to become negative; note that the overall clot still becomes compliant while at the same time intra-cycle stiffening also takes place. Subregion 2c: G´ increases, J’_M_ increases whereas J’_L_ decreases. Region 2 ends at the J’_M_ maximum. Region 3: macroscopic shear stiffening response: G´ increases; J’_M_ and J’_L_ decrease. This region ends at the J’_L_ and J’_M_ minima. Region 4: yielding and breakup: G´ decreases, J’_L_ and J’_M_ increase again, the clot breaks. (**b,d,f**)**:** Test regions when blood cells are absent (e.g., in PDP and fibrinogen solutions). Region 1: LVE region (G´ is constant, J’_M_ = J’_L_). Region 2: G´ increases and the compliances decrease but remain converged. Region 2 ends at the divergence of the compliances (J’_M_ > J’_L_). Region 3: G´ increases, J’_M_ and J’_L_ decrease, *R* < 0. Region 4: G´ decreases, J’_M_ and J’_L_ increase; the clot breaks.

**Figure 8 molecules-25-03890-f008:**
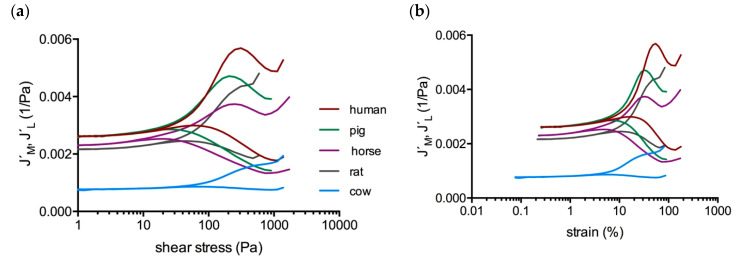
Compliances of clots at 25 G/L platelet count (rounded value) against (**a**)**:** Stress and (**b**)**:** strain. Human clots (ID: 01, brown curves) exhibit the highest, and cow clots the lowest compliances among these species. In this illustration the pig clot (green curves) is the most comparable one to the human clot. Cow takes an exceptional position: clots gain much lower strains and show a shift of the thresholds to higher stresses. Like the cow clot, also the rat clot (grey curves) tends to display a J´_M_ shoulder instead of a maximum.

**Figure 9 molecules-25-03890-f009:**
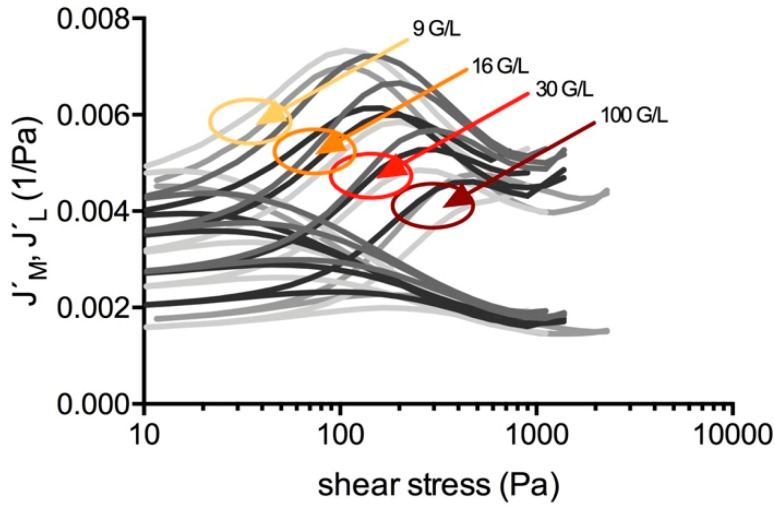
Compliance curves of clots from the plasma samples of three human volunteers (ID: 01– 03) having various platelet counts. The platelet count (rounded values for simplicity) determines the shape of the compliance curves.

**Figure 10 molecules-25-03890-f010:**
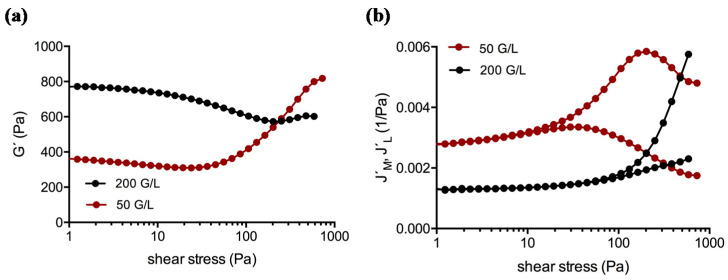
(**a**)**:** G´-values and (**b**)**:** compliances of human plasma samples (ID: 02). If the platelet count is too high (black curves), clots lose the ability to shear-stiffen and break earlier. For simplicity, the given platelet counts reflect rounded values.

**Table 1 molecules-25-03890-t001:** Platelet count (PLT) of platelet-rich plasma used for preparing the dilutions and plasma fibrinogen concentration (FIB) of the samples used.

Individual	PLT (G/L)	FIB (g/L)
Human: 01	280	2.51
Human: 02	748	3.05
Human: 03	124	3.18
Cow	794	2.64
Pig	218	1.91
Rat	162	2.61
Horse	447	2.19

**Table 2 molecules-25-03890-t002:** Rheometry parameters.

Parameter	Description
G´	Shear elastic modulus; reflects clot stiffness.
G´_plateau_	Maximum stiffness that the clot reached during its generation. Reflects also the maximal clot stiffness at “rest”—this means at equilibrium conditions. Here, elastic behavior (reversible deformation) takes place.
γ(ω)	Cyclic strain, clot deformation during sinusoidal oscillation at a certain angular frequency. At a given shear stress, a stiff material will deform less than a soft material.
J’_M_	Minimum strain compliance. Compliance of the clot while the oscillation amplitude crosses the zero point (part of the cycle, where the clot is only minimally stretched out). A stiff material has a low compliance because it responds to the shear stress with only a small strain increment. As a result, the tangent through the zero point has a low gradient.
J’_L_	Large strain compliance. Compliance of the clot at the highest stress value during the oscillation cycle. Here, the clot is maximally strained (= stretched out) within the Large strain compliance. Compliance of the clot at the highest stress value during the oscillation cycle. Here, the clot is maximally strained (= stretched out) within the respective cycle. Likewise, a stiff material has a low compliance, and the tangent from the zero point to the maximum strain of the clot has therefore a low gradient.
*R*	Stress softening ratio: relationship between J’_M_ and J’_L_. If *R* = 0, both compliances are equal and the clot experiences elastic deformation (the clot is in its equilibrium). If *R* ≠ 0, the clot experiences plasticity that originates from cyclic loading.
τ_D_	Shear stress at which J’_M_ and J’_L_ diverge. At this point plasticity sets in that originates from cyclic stretching. The clot cannot relax into its original state beyond this critical shear stress.
τ_L_	Shear stress needed for the J’_L_-maximum. Here, the clot is most compliant at its maximal stretch-out during the oscillation cycle. After this threshold, the clot becomes stiffer.
τ_M_	More shear stress is needed to reach the J’_M_-maximum, because here the clot is probed when the oscillation amplitude crosses the zero point (the clot is only minimally stretched out within the cycle). τ_M_ is therefore the shear stress threshold to gain the best clot compliance during that part of deformation. Likewise, after this threshold, the clot becomes stiffer.
G´_max_	Maximum stiffness that a clot can gain before damage.
γ(ω)_max_	Maximum deformation that a clot can gain before damage.

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
