# Peer review of "Blood Clot Phenotyping by Rheometry: Platelets and Fibrinogen Chemistry Affect Stress-Softening and -Stiffening at Large Oscillation Amplitude"

_molecules, 2020, doi:10.3390/molecules25173890_

Round 1
Reviewer 1 Report
The manuscript is an in vitro rheological study on coagulated blood clots, with focus on the regimes and their onset limits where deviations from the inear elastic model occurs. The clot behavior from healthy volunteers and four mammal specii were compared. Blood smaples were pre-treated by centrifugation. The "model" clots consisted of the fibrinogen network plus platelets in variing ratio, but no other blood components. The tests were carried out on a high-sensitivity, state-of-the-art rheometer, in dynamical mode. The results are discussed on the basis of clot network structure and the effect of the platelet amount within.
The Introduction section is comprehensive and understandable for non-specialists. It describes differeneces between arterial and venous blood coagulation, differences between clot structure and composition, homogeneity, the roles of different blood components within the clot and their influence if the clot is under shear. Rheometry techniques, typically used for such investigations, and their strengths and limitaions are discussed.
The results are presented in a clear way, the figures are appealeing in appearance, and the measurements are meticulous. The discussion section is thorough, the rheology - network structure relationships are analyzed, and hypotheses behind the non-affine/affine deformation regimes are plausible. The language is pleasent to read and the typing is accurate. The conclusion confirms that the investigation of the nonlinear part of the rheological behavior is essential, because this is the regime where correlation with the composition (platelet ratio) within the blood clot strikingly reveals, can therefore give useful information on a patient`s coagulation conditions.
I recommend the manuscript for acceptance after minor revision.
Questions:
Q1
Was the blood coagulation process, so the formation of the "model" clots, very repeatable? The Statistics section (4.4) should include some information about it, e.g. if the stress runs were repeated on newly diluted samples, or if the amount of sample blood (after "fractioning" by centrifugation) was sufficient for several tests.
Q2
The 3 human individuals had quite different platelet numbers according to Table 1. However, all subsequent figures (except Fig.9) show only 1 human sample. Which sample is shown in these figures? Is it an average (mixture)? I do not find any hint on it under the Methods.
Q3
The multiple fibres adhering to incorporated platelets reminds me on the structure of so-called Laponite gels. Laponites are nanometer-sized synthetic silicate disks (sythetic clay), and they can be used as cross-linkers in free-radical polymerization. Due to the high functionality of these super-crosslinkers, Laponite gels are famous for their exceptional high extent of stretching, easily up to 1000%, without rupture. Is there an "optimum" platelet amount in the blood clot? Are there cases when a high or low onset, respectively, of Region 4 (clot break) is more favorable?
Remarks:
R1
Lines 116-122: I think this paragraph belongs rather to the conclusion. Here, at the end of the introduction, I would repeat the questions once more that are expected to be answered by the study.
R2
Line 155: Change names to ref. numbers
R3
Letters a,b,c,d,etc. are missing on some of the figures, Fig.1,6,7,10.
R4
Fig.2d: Could the orange symbols have another form (e.g. down-triangles) to see them behind the brown up-triangles?
Author Response
Q1 Was the blood coagulation process, so the formation of the "model" clots, very repeatable? The Statistics section (4.4) should include some information about it, e.g. if the stress runs were repeated on newly diluted samples, or if the amount of sample blood (after "fractioning" by centrifugation) was sufficient for several tests. We were unable to perform technical repeats due to the time needed to run one complete test (kinetic+LAOStress-test+preparation for the next run = about 75 minutes). We were afraid that the platelets could deteriorate meanwhile. A blood collection on a subsequent day from the same individual will not provide samples suitable for a repeated measurement, since previous experiments showed us that on the subsequent day clot stiffness can be changed and the sample must be regarded as a different sample. In order to perform as many dilutions as possible with the available PRP and PDP volumes from one volunteer, we decided to perform only one run per dilution and validate the findings like it is shown in figure 9. We recently performed technical triplicates in PDP samples incubated with LPS versus vehicle and found good repeatability (please see doi: 10.3389/fimmu.2020.01551). Please see also the figure below, showing three technical repeats for each condition; grey: PDP with vehicle, black: PDP with LPS in vehicle: To clarify this in the manuscript, we included a more precise description of how we prepared the dilutions in section 4.2 (line 467ff) and about repeatability in the statistics section 4.4 (line 505ff). Q2 The 3 human individuals had quite different platelet numbers according to Table 1. However, all subsequent figures (except Fig.9) show only 1 human sample. Which sample is shown in these figures? Is it an average (mixture)? I do not find any hint on it under the Methods. For clarity, figures 2-8 show results from the human individual #1 only. This is now indicated in the respective figure legends. Figure 9 shows the behaviour at large sinusoidal amplitude of all three human individuals and therefore the repeatability of the tests. Figure 10 shows data from the human individual #2, because the PRP of this individual had the highest platelet count. The origin of the samples is now indicated in all figure legends. Q3 The multiple fibres adhering to incorporated platelets reminds me on the structure of so-called Laponite gels. Laponites are nanometer-sized synthetic silicate disks (synthetic clay), and they can be used as cross-linkers in free-radical polymerization. Due to the high functionality of these super-crosslinkers, Laponite gels are famous for their exceptional high extent of stretching, easily up to 1000%, without rupture. Is there an "optimum" platelet amount in the blood clot? Are there cases when a high or low onset, respectively, of Region 4 (clot break) is more favorable? Laponite gels are indeed quite fascinating, but there are certainly differences between them and blood clots. In Laponite gels, the discs are an integral part of the network. The structure strength is given by the interaction forces between the discs. In blood clots the network is mainly formed by fibrin fibers and the platelets (resembling discs) adhere to the fiber network and are distributed with the network. From a rheological perspective, Laponite gels can have very long linear viscoelastic regimes, but they do not show any stress stiffening as the fibrin networks with or without platelets can do. Stress-stiffening must therefore be a quality of extensible fibers, while the adhering or incorporated discs only shift certain thresholds but can also block fiber extension if there are too many discs (= platelets) present. The platelets are influencing the rheological responses mainly at the lower stress levels. More platelets make the clots stiffer. Concerning the question about an optimal platelet count: platelets are needed for clot formation and they enhance clot stiffness, but they reduce the stiffening response of the fiber meshwork. Amazingly, even if the fiber system contains very few platelets and has therefore a low stiffness at LVE, the load that this clot can bear is equivalent to the load that a stiffer clot with a higher platelet count can withstand. The prize to pay for a low platelet count is only a small retardation of clotting time (see figure 1). In contrast, platelet-fiber clusters are stiff but brittle and cannot uptake the same cyclic loading. They form compact clusters. Figure 10 shows that a platelet count of 200 G/L already blocked the stiffening response (a platelet count of 200 G/L is within the physiological range). Looking on these results, I would advice that clinicians should keep the platelet count of their patients low and not try to raise it up to a median physiological value. This is generally done in the clinics. Platelet concentrates are transfused only at very low platelet counts of patients, because platelets are so effective and clinicians are afraid of thrombosis. It is the fibrinogen concentration that is more balanced in patients postoperatively by using fibrinogen concentrates. There are no cases, where a clot breakup at lower loads is more favourable. But clots should be ductile to allow remodelling. Remodelling is achieved by shear forces (outer clot layers) and by plasminogen factors that must penetrate into the clot against the simultaneous tendency to be washed away by the bloodstream. A compact cluster of any composition (platelet-vWF clusters, platelet-fibrin clusters, RBC aggregates) will reduce the diffusion of these factors into the clot. Again, a low platelet count will be favourable. We included a sentence about the relevance to keep the platelet count low into the discussion section of the revised manuscript (line 425ff). Remarks: R1 Lines 116-122: I think this paragraph belongs rather to the conclusion. Here, at the end of the introduction, I would repeat the questions once more that are expected to be answered by the study. We modified and shortened this paragraph R2 Line 155: Change names to ref. numbers Done. Thank you R3 Letters a,b,c,d,etc. are missing on some of the figures, Fig.1,6,7,10. Has been corrected R4 Fig.2d: Could the orange symbols have another form (e.g. down-triangles) to see them behind the brown up-triangles? Like in the other three graphs of this figure, the symbols are now circles, but we enlarged the orange symbols so that the difference/overlap between orange and brown can be better made out.

Reviewer 2 Report
The authors performed viscoelastic tests to determine the influence of the platelet count on the network level in clots. They used a stress amplitude sweep test (LAOStress) to investigate clots from plasma samples diluted to five different platelet concentrations. Five species were used to validate the protocol (human, cow, pig, rat, horse). For all species except cow, they found that cyclic stress loading generates a characteristic strain response that is dependent on the platelet count only at low stress. They conclude that their protocol provides several thresholds to connect the softening and stiffening behavior of clots and therefore opens a new route to describe a blood clot´s phenotype.
This is an interesting manuscript and the method could find applications in diagnostics.
Nonetheless, a couple of modifications would be useful:
- 1: Are the curves for the human sample from one individual and if so which? This is important to know since human 2 has a pathologically high plt count. Where are the curves for the other two human samples?
Or are these the mean curves for all three human samples? In that case, the error bars are missing.
- In all experiments: how many independent experiments were performed? Has always just one measurement per sample been performed? If so why?
- Why was the plt count in human 2 so high? Was it caused by the PRP preparation?
- The authors only discuss the influence of fibrinogen but von Willebrand factor (VWF) is abundant in plasma too and it is a force responsive protein which is most likely to highly influence the rheology of clots especially at high stresses! VWF depleted human plasma should also be tested to investigate the contribution of VWF to clot stiffness at different stresses.
- The authors claim that their protocol could “open a new route to describe a blood clot´s phenotype”. Please discuss the physiological applications, for example, in patients with thrombocytopenia or high platelet counts!
- For non-expert readers, the manuscript is quite difficult to follow. To improve readability it would be helpful if the authors would explain a bit more what each parameter means. What clot characteristic can be explained by which parameter and what conclusions can be drawn from the data which respect to the clot behavior.
- Minor point: In Figs.1, 6, 7, 8, 10 the panel names (a,b,c etc.) are missing.
Author Response
- Are the curves for the human sample from one individual and if so which? This is important to know since human 2 has a pathologically high plt count. Where are the curves for the other two human samples?
Or are these the mean curves for all three human samples? In that case, the error bars are missing.
For clarity, figures 2-8 show results from the human individual #1 only. This is now indicated in the figure legends. Figure 9 shows the compliances of all three human individuals and therefore the repeatability of the tests. Figure 10 shows data from the human individual #2, because the PRP of this individual had the highest platelet count.
The curves of the other two human individuals are below:
Adding these graphs e.g. as Supportive Information do not add further value to the main finding but would more confuse the reader. Figure 9 already shows the behaviour at large sinusoidal amplitude of all human individuals.
- In all experiments: how many independent experiments were performed? Has always just one measurement per sample been performed? If so why?
PRP and PDP of one individual were mixed together on the day of blood withdrawal and all tests had to be finalized on the same day due to sample ageing (5 tests plus 1 test with whole blood while the main portion of blood was centrifuged à about 75 minutes for each test). Technical duplicates were impossible due to potential ageing of PRP. Repeatability of our approach is shown in figure 9 and can be accesses from a recent study (doi: 10.3389/fimmu.2020.01551). All sample preparation and measurements were done by only one person to avoid bias due to different handling procedures.
- Why was the plt count in human 2 so high? Was it caused by the PRP preparation?
Platelet count in the human individual #2 was so high because the whole blood of this volunteer had a higher platelet count (see the table below). All individuals were clinically healthy and fulfilled the inclusion criteria given in the M&M section. We added a remark on the variability of PRP in lines 471-472.
|
ID |
whole blood |
||
|
human |
RBC (T/L) |
WBC (G/L) |
PLT (G/L) |
|
#1 |
4.1 |
6.51 |
79 |
|
#2 |
4.7 |
8.23 |
217 |
|
#3 |
4.8 |
5.43 |
45 |
- The authors only discuss the influence of fibrinogen but von Willebrand factor (VWF) is abundant in plasma too and it is a force responsive protein, which is most likely to highly influence the rheology of clots especially at high stresses! VWF depleted human plasma should also be tested to investigate the contribution of VWF to clot stiffness at different stresses.
We thank the reviewer for this suggestion. Circulating vWF unfolds into a linear conformation at high shear, captures platelets from the bloodstream, protects FVIII from clearance, and generates a fiber network in proximity to the vessel wall. vWF has become an important clinical marker during the past years. It is mainly incorporated into arterial clots and will certainly influence their performance like many other factors that are present in plasma (albumin binds to the fibers and affects their mobility against each other, the sum of plasma proteins enhances matrix viscosity). vWF-platelet assemblies will develop predominantly in clot layers closer to the arterial wall. Since our approach works not only in fibrin networks, but in other biological networks as well (we have few preliminary data on collagen), we expect that we can identify the stress thresholds also for vWF networks with and without platelets in a subsequent study. In this study, vWF-platelet clusters will not play a dominant role because we generated the clots at very low strain, not high enough to unfold vWF.
- The authors claim that their protocol could “open a new route to describe a blood clot´s phenotype”. Please discuss the physiological applications, for example, in patients with thrombocytopenia or high platelet counts!
This suggestion was added into the discussion section (see line 425ff).
- For non-expert readers, the manuscript is quite difficult to follow. To improve readability it would be helpful if the authors would explain a bit more what each parameter means. What clot characteristic can be explained by which parameter and what conclusions can be drawn from the data, which respect to the clot behavior.
A table that explains the parameters has been added (à table 2 in section 4).
- Minor point: In Figs.1, 6, 7, 8, 10 the panel names (a,b,c etc.) are missing.
Has been corrected.

Round 2
Reviewer 2 Report
All comments have adequately been addressed.